# Association Between Decision-Making Styles, Personality Traits, and Socio-Demographic Factors in Women Choosing Voluntary Pregnancy Termination: A Cross-Sectional Study

**DOI:** 10.3390/ejihpe15100214

**Published:** 2025-10-16

**Authors:** Letizia Lorusso, Nicola Bartolomeo, Maria Elvira Metta, Daphne Gasparre, Patrizia Pignataro, Giulia Caradonna, Paolo Taurisano, Paolo Trerotoli

**Affiliations:** 1School of Medical Statistics and Biometry, Interdisciplinary Department of Medicine, University of Bari Aldo Moro, Piazza Giulio Cesare 11, 70124 Bari, Italy; mettamariaelvira@gmail.com; 2Interdisciplinary Department of Medicine, University of Bari Aldo Moro, Piazza Giulio Cesare 11, 70124 Bari, Italy; paolo.trerotoli@uniba.it; 3Department of Translational Biomedicine and Neuroscience (DiBraiN), University of Bari Aldo Moro, Piazza Giulio Cesare 11, 70124 Bari, Italy; daphne.gasparre@uniba.it (D.G.); patrizia.pignataro@uniba.it (P.P.); paolo.taurisano@uniba.it (P.T.); 4U.O.S.V.D. Pianificazione Familiare “Di Venere Fallacara”, Triggiano, 70131 Bari, Italy; giulia.caradonna@asl.bari.it

**Keywords:** women health, voluntary termination of pregnancy, personality trait, decision-making style, psychological factors, socio-demographic factors, educational level

## Abstract

Personality traits, decision-making styles, and socio-demographic information help understand the characteristics of women requesting voluntary termination of pregnancy (VTP) and monitor the timing and quality of this service However, our current data do not allow for an in-depth exploration of additional, less evident factors that might influence the decision. The aim was to characterise a sample of women requesting VTP in terms of socio-demographic, personality, and decision-making style variables, and to examine the associations among these factors. We conducted a cross-sectional study administering the General Decision-Making Style (GDMS) test, the Big Five Questionnaire (BFQ) and the Personality Inventory for DSM-5 (PID-5) to women requesting VTP certification. BFQ scores were generally high, especially for “Conscientiousness” and “Openness”. “Agreeableness” and “Openness” scores were lower in women with low vs. high education (diff = −8.2 [−13.9, −2.4] and diff = −7.5 [−15.0, −0.1]); “Extraversion” was higher in employed women. PID-5 scores for “Detachment” and “Psychoticism” were higher in medium vs. high education (diff = 1.6 [0.05, 3.2] and diff = 1.9 [0.1, 3.8]), but not at pathological level; “Negative affect” was lower in women with children (diff = −1.6 [−3.1, −0.2]). GDMS scores were not associated with socio-demographic factors. “Avoidant” and “Spontaneous” styles were negatively associated with “Conscientiousness” and “Emotional Stability”; “Rational” style was positively associated with “Conscientiousness” and “Disinhibition”. High BFQ and low PID-5 scores suggest no personality dysfunction in women undergoing VTP. No predominant decision-making style emerged, but associations between personality traits, decision-making, and socio-demographic factors such as educational level, employment and parity, were observed.

## 1. Introduction

In Italy, the voluntary termination of pregnancy (VTP) is regulated by law 194, a law issued by parliament on the 22 May 1978 ([33]). On the 29 July 1975, law 405 established family planning counselling facilities for the administration of necessary means to allow freely chosen decisions to be made regarding responsible procreation, including for minors ([18]; [33]; [34]). The first objective of law 194 was to protect the rights of women to access a path of care from conception.

This pathway entails first contact with family planning facilities, which guide women in choosing a gynaecology and obstetrics operating unit at an accredited public or private healthcare service. The epidemiological surveillance system related to the VTP has been active since 1980. It was included in the national surveillance system on the 3 March 2017 ([18]; [26]) with the aim of monitoring the health of mothers and children and gathering important information about the socio-demographic characteristics of women who choose VTP and the main characteristics of the corresponding intervention ([18]).

While research has extensively addressed the medical and socio-demographic aspects of abortion, there remains limited understanding of how psychological dimensions—such as personality traits and decision-making styles—interact with women’s experiences of VTP. Recent studies have emphasised the role of internal psychological factors, including perceived self-efficacy, emotional resilience, and personality functioning, in shaping abortion-related decision-making and post-abortion adjustment ([7]; [39]; [43]).

Decision-making in the context of abortion is often influenced by a complex interplay of emotional, social, cultural, and economic factors ([4]; [8]; [20]; [44]; [46]; [53]). Abortion stigma, for instance, may limit emotional expression and increase decision-related stress, especially in socially conservative environments. Factors such as socioeconomic status, education, and employment can further impact women’s autonomy in reproductive decisions ([2]; [3]; [21]). These dynamics underline the importance of understanding psychological profiles beyond psychopathological risk.

Recent literature also suggests that individual differences in cognitive style and personality traits may influence how decisions about abortion are experienced and justified ([13]; [37]). However, few studies have systematically investigated the relationship between established personality models, such as the Big Five, or maladaptive personality traits (as defined by DSM-5) and decision-making styles in the context of VTP.

Although various psychometric tools have been used to explore personality traits in clinical and general populations, findings related to women requesting VTP remain fragmented and often lack standardisation across studies. Most existing studies are limited by small samples, heterogeneous methodologies, or lack of integration between psychological and socio-demographic data ([20]).

The present study aims to contribute to this underexplored area by investigating the associations between personality traits, decision-making styles, and socio-demographic variables in women requesting VTP certification. We administered the General Decision-Making Style (GDMS) test ([25]), the Big Five Questionnaire (BFQ) ([11]), and the Personality Inventory for DSM-5 (PID-5) ([22]). Understanding the interplay between psychological and demographic factors can support tailored counselling interventions and help improve the quality and accessibility of reproductive health services.

## 2. Materials and Methods

### 2.1. Participants and Procedure

A cross-sectional, observational study was conducted on women aged 18 years or older who underwent voluntary termination of pregnancy at the Family Planning Unit of the “Di Venere-Fallacara” hospital in Triggiano between November 2023 and January 2024. The study protocol was reviewed and approved by the local ethics committee. This study was conducted in accordance with the ethical principles of the Declaration of Helsinki. The women who agreed to participate were asked to provide informed consent at the time of requesting the certification required for the VTP or at the time of the procedure itself. Data were collected anonymously to safeguard the privacy of the participants. A unique identification code unrelated to personal or sensitive data was assigned to each patient. Personal data processing was carried out according to the principles of fairness, lawfulness, and transparency, pursuant to Article 24 of Regulation (EU) 2016/679.

In Apulia, from 1 November 2023 to 31 January 2024, a total of 1210 VTPs were carried out. At the Family Planning Unit of the “Di Venere-Fallacara” hospital in Triggiano, 198 VTPs were conducted. Of these cases, 74% corresponded to women who provided informed consent to participate voluntarily in this study. However, two of these participants withdrew. After excluding data with missing values, 62% of the total participants were included in the analyses, as shown in the flowchart in Figure 1.

The women were administered a questionnaire divided into two sections. The first section included date of birth, province of birth and residence, municipality of birth and residence, citizenship, marital status, educational level, employment status and professional position, economic activity sector, number of previous pregnancies (live births, stillbirths, previous spontaneous abortions, and voluntary terminations of pregnancy), weeks of amenorrhea, the presence of foetal malformations, the date of the procedure, the date of certification, the facility that issued the certification, any urgency, where the voluntary termination of pregnancy (VTP) was carried out, the type of procedure, pain management, details on the care pathway (the type of hospitalisation and the number of days/accesses), and complications.

The second section included tests for assessing personality traits and decision-making styles, complete with instructions. The tests were self-administered, and a physician trained in biostatistics was available to assist in the collection of data if requested by nurses or the participants. The average time required to complete the questionnaire was 40 min.

### 2.2. Assessment of Personality Traits and Decision-Making Style

The following three psychological tests were administered; in these tests, each item was structured as a statement. The patients were then asked to evaluate how accurately each statement represented them, responding based on a Likert scale. We calculated Cronbach’s alpha to evaluate the internal consistency of the items for each dimension of the three tests.

The Big Five Questionnaire (BFQ) is a widely used psychometric tool designed to assess personality traits through five broad dimensions: energy (extraversion), agreeableness, conscientiousness, emotional stability (neuroticism), and openness (openness to experience). The BFQ consists of 132 items, with each item rated on a 5-point Likert scale from 1 “completely false for me” to 5 “completely true for me”. It is validated for use in various cultural contexts and has demonstrated robust psychometric properties, including high reliability and validity. In our sample, all BFQ subscales showed acceptable internal consistency, with Cronbach’s alpha coefficients above 0.70, in line with the original validation study, where alpha values ranged from 0.73 to 0.90 across the five dimensions ([11]).

The Personality Inventory of the Diagnostic and Statistical Manual of Mental Disorders, Fifth Edition, Brief Form (PID-5-BF), comprises 25 items, each rated on a 4-point Likert scale from 0 “Always or often false” to 3 “always or often true”. It evaluates five broad domains of maladaptive personality traits: negative affect, detachment, antagonism, disinhibition, and psychoticism. In our sample, Cronbach’s alpha coefficients for the PID-5 dimensions ranged from 0.70 to 0.77, indicating acceptable internal consistency and aligning with previous Italian validation studies reporting alpha values generally above 0.80 ([22]).

The General Decision-Making Style (GDMS) is a psychometric tool used to assess an individual’s preferred approach to making decisions. It evaluates five distinct decision-making styles: rational, intuitive, dependent, avoidant, and spontaneous. Each style represents a different method with which individuals process information and come to decisions. The GDMS consists of a series of statements ranked on a 5-point Likert scale ranging from 1 “strongly disagree” to 5 “strongly agree.” This inventory is widely used in organisational psychology for its ability to provide insights into the cognitive and behavioural patterns that underline decision-making. In our sample, the internal consistency of the GDMS subscales was acceptable, with Cronbach’s alpha values ranging from 0.68 to 0.81, consistent with prior validation studies reporting coefficients between 0.62 and 0.94 ([25]; [45]).

### 2.3. Statistical Analysis

For the analysis, marital status was dichotomised into two categories: “in a couple” (married or in a civil partnership) and “not in a couple” (single, widowed, separated, or divorced). Age was divided into two groups: “<30 years” and “≥30 years”. Based on the “parity” variable, we distinguished between women with children and those without. Employment status was categorised into two groups, namely, “employed” and “unemployed,” which included the categories “unemployed”, “seeking first employment”, “housewives”, and “students”. Educational level was divided into three groups: “low education” (corresponding to an elementary school certificate or a lower-secondary-school certificate), “medium education” (corresponding to an upper-secondary-school diploma), and “high education” (corresponding to a university degree or other higher-education qualifications). Weeks of amenorrhea were categorised into “<9 weeks” and “≥9 weeks”.

To assess whether the sample examined in the study was biassed due to monocentric recruitment, chi-squared tests were used to compare the socio-demographic characteristics and the characteristics of the VTP event of the sample with those of the entire population of women who underwent VTP in the Puglia region during the same period.

For each questionnaire’s score, the mean and standard deviation or the median and interquartile range were calculated based on a normality check, which was performed using the Shapiro–Wilk test. To assess associations both within the dimensions of each individual test and between tests, cross-correlations among all dimensions of the three psychological tests were measured using Pearson’s parametric correlation coefficient or Spearman’s non-parametric correlation coefficient, selected based on the normality assumptions. Bivariate normality of the joint distribution of the variables was verified using Mardia’s test for multivariate normality.

First, multivariable analyses of covariance (ANCOVA, within the GLM framework) was conducted to explore the associations between the individual dimensions of the BFQ and PID-5 with socio-demographic characteristics (age, marital status, education, employment status, and parity). Results are reported as estimated mean differences with 95% confidence limits. Subsequently, univariate and multivariable general linear regressions were performed to investigate the associations between GDMS outcome scores, i.e., decision-making styles, the scores for the BFQ, and the PID-5 scores and socio-demographic variables. In building multivariable regression models, we perform univariate screening of potential predictors, selecting those with a liberal *p*-value threshold (using *p* < 0.25 helps avoid this under-specification risk) for consideration in the multivariable model ([10]). The rationale is that using a stricter threshold (e.g., *p* < 0.05) at the univariate stage may prematurely exclude variables that do not show strong associations alone but become significant in combination with other covariates or act as confounders. To address potential concerns regarding multicollinearity in the multivariable models, we calculated collinearity diagnostics (Variance Inflation Factors and tolerance values) for all predictors. The diagnostics indicated acceptable values (maximum VIF = 2.83; minimum tolerance = 0.35), confirming the absence of problematic collinearity. The pairwise multiple comparisons were adjusted using Tukey’s test. The results of the regressions, presented through regression coefficients and their standard error (SE) or 95% confidence levels, were discussed to provide a comprehensive overview of the relationship dynamics among the variables involved.

Results with a *p*-value < 0.05 were considered statistically significant. The statistical analysis was conducted using SAS/STAT^®^ Software, version 9.4.

## 3. Results

We recruited 122 women for analysis. Their overall age range was between 18 and 46, with an average of 31.8 (and an SD of 7.51). The most represented age range was between 30 and 34 years (23%), while the least represented was that between 18 and 19 years (8.2%). A total of 94.3% of the patients were Italian citizens. In total, 58.2% of the women had a medium educational level, and 54.6% were employed. A total of 52.5% of them had at least one child, while the percentages of those who had suffered previous spontaneous abortions and who had previously undergone VTP were 13.9% and 27.8%, respectively. In total, 88.1% of the VTPs had been carried out within the 8th week of amenorrhea. Altogether, 4.1% of the cases corresponded to foetal malformation, and only for 26.2% was the VTP declared urgent. A total of 99.2% of the VTPs were pharmacological (Table 1).

None of the results of the chi-squared tests, conducted to compare the socio-demographic characteristics of our sample with the characteristics of all women who underwent VTP in the Apulian region during the same period, were statistically significant.

The highest mean score in the BFQ test was in the dimension “Conscientiousness”. The highest dimension score in the PID-5 test was for “Negative Affect”. Overall, the PID-5-dimension scores were low, and none of the selected women showed a pathological score. The higher dimension scores in the GDMS test were “Rational”, “Intuitive”, and “Dependent” (Table 2).

Figure 2 shows the intra- and inter-dimensional correlations between the questionnaires. The dimensions of the BFQ had a positive and statistically significant internal correlation, except for “Emotional stability,” which was not significant correlated with other dimensions. The internal correlations between the dimensions of the PID-5 questionnaire were stronger and almost always statistically significant. The internal correlations in the GDMS questionnaire were generally positive, but the dimension “Rational” had a significantly negative correlation with “Spontaneous”.

The correlations between the dimensions of the BFQ and the dimensions of the PID-5 were almost always negative and, in many cases, statistically significant, except for “Extraversion” and “Emotional Stability,” which showed negative and statistically significant correlations, respectively, with the PID-5 dimensions “Detachment” and “Psychoticism”; and “Disinhibition” and “Negative Affect”. Figure 2 shows very low to low but statistically significant correlations (ranging from –0.25 to –0.47) between all PID-5 domains and Agreeableness; similarly, very low to low and significant negative correlations (from –0.27 to –0.39) were found between all PID-5 domains and Emotional Stability. Thus, since the BFQ traits are overall negatively correlated with the PID-5 domains, elevated scores in BFQ should not be taken as indicators of dysfunction. The “Avoidant” and “Spontaneous” scores for decision style tended to increase as the PID-5 dimensions score increased; otherwise, they tended to decrease as the BFQ dimensions “Agreeableness”, “Conscientiousness”, and “Emotional Stability” increased. The dimension “Rational” had a significantly positive correlation with the BFQ dimensions “Conscientiousness” and a negative correlation with “Disinhibition”.

### 3.1. Associations Between Big Five Questionnaire and Socio-Demographics Characteristics

In the multivariable analysis of covariance, the scores for the “Agreeableness” dimension were significantly lower for women with low educational levels compared to those with high levels (mean diff = −8.2, 95%; CI = −13.9 to −2.4). The scores for the “Openness” dimension were also significantly lower for women with a low educational level compared to those with high levels (mean diff = −8.2; 95% CI = −15.4 to −0.9). The scores for the “Extraversion” dimension were significantly higher among employed women than unemployed women (mean diff = 3.7, 95% CI = 0.04 to 7.4), while the association for the scores of the “Emotional Stability” dimension was not statistically significant (Figure 3). The complete analysis is provided in the Appendix A.

### 3.2. Associations Between PID-5 Questionnaire and Socio-Demographics Characteristics

In the multivariable analysis of covariance, the “Detachment” score remained associated with educational level, with the scores being significantly higher for the women with low (mean diff = 2.3, 95% CI = 0.24 to 4.32) and medium (mean diff = 1.6, 95% CI = 0.05 to 3.2) educational levels compared to those for the women with high educational levels. Additionally, for the dimension “Psychoticism” the scores were significantly higher among women with medium educational levels compared to those for the women with high educational levels (mean diff = 1.9, 95% CI = 0.06 to 3.8). The scores for the dimension “Negative affect” were significantly lower for women with children compared to those for women without children (mean diff = −1.6, 95% CI = −3.1 to −0.2). The dimensions “Antagonism” and “Disinhibition” did not show any statistically significant associations with socio-demographic variables (Figure 4). The complete analysis is provided in the Appendix A.

### 3.3. Associations Between General Decision-Making Style Questionnaire, Socio-Demographic Characteristics, and the Dimensions of the BFQ and PID-5 Tests

In the multivariable regression, the “Avoidant” score maintained a statistically significant relationship with the dimensions “Conscientiousness” (β = −0.16; 95% CI = −0.24 to −0.07) and “Emotional Stability” (β = −0.07; 95% CI = −0.13 to −0.003). The “Rational” decision style was significantly associated with “Conscientiousness” (β = 0.14; 95% CI = 0.07 to 0.21) and “Disinhibition” (β = −0.58; 95% CI = −0.86 to −0.30). The scores for “Spontaneous” only maintained a statistically significant association with the dimensions “Conscientiousness” (β = −0.12; 95% CI = −0.21 to −0.04), “Extraversion” (β = 0.18; 95% CI = 0.09 to 0.27), and “Emotional Stability” (β = −0.07; 95% CI = −0.13 to −0.01) (Figure 5). The complete analysis is provided in the Appendix A.

## 4. Discussion

In this cross-sectional study a population of women undergoing VTP participated in the study to investigate the association between socio-demographic characteristics, personality traits, and decision-making abilities evaluated by BFQ, PID-5, and GDSM questionnaires. We may exclude a possible selection bias due to the monocentric nature of this survey because the socio-demographic characteristics of participating women did not differ from those who underwent a VTP in the Puglia region in the same time period.

The choice of women who seek VTP may depend on various situations determined by a person’s social context. In our sample, the most represented socio-demographic conditions among women requesting VTP were being employed, being in a relationship, having a medium level of education, and having children. [20] ([20]) similarly reported that employment status and relationship status (e.g., single/divorced) were frequently observed among women who choose VTP, although they did do not provide evidence concerning educational level or parity. On the contrary, [7] ([7]) in U.S. sample of women with unintended pregnancies, found that those who decide to terminate were more likely to have lower levels of education, lower household income, and to be single or divorced, when compared with women who continue their pregnancies. The average scores obtained through the BFQ personality test are generally high for all traits, but “Conscientiousness” and “Openness” stand out among the various dimensions. These characteristics are typical of individuals who are meticulous and persevering, with a great openness to new experiences and a deep appreciation for culture ([29]; [50]) However, it is important to acknowledge that these ‘personality’ traits are not solely individual qualities but are also shaped by external factors such as socio-cultural context, economic conditions, and life experiences ([17]; [42]). For example, women with different characteristics in terms of age, marital status, education level, employment, and parity showed different nuances in the five dimensions of the Big Five.

Numerous longitudinal studies have demonstrated that personality traits may be affected by significant (major) life events ([16]; [32]; [52]), and that the accumulation of more than one (minor) such event can mediate substantial changes in personality traits. However, experts have not reached complete consensus regarding the directionality of these changes ([52]). However, our cross-sectional and observational design did not allow us to assess changes in personality traits before, during, and after the decision to seek voluntary termination of pregnancy (VTP). In our sample, we assumed that women had already made their decision, and that the life event—realisation of an unwanted pregnancy—had only recently occurred. Thus, we were measuring the personality traits as they manifested at that particular moment when the decision for VTP was taken.

It is plausible that sociodemographic characteristics preceding the measurement of personality traits may influence the expression of those traits. Empirical research has found that variables such as age, gender, educational attainment, socioeconomic status, and employment status are significantly associated with levels of diverse personality dimensions ([28]; [36]). Consequently, when interpreting trait scores, we considered the antecedent sociodemographic context as potentially shaping those scores, focusing specifically on the associations between sociodemographic factors and personality traits. For example, [14] ([14]) investigated the effect of external and internal sources on personality traits, highlighting how new life experiences may affect personality domains as “Extraversion” and “Openness”. In our study, a higher score in the “Extraversion” domain was observed among women with employment compared to unemployed women.

A previous longitudinal study that followed 5672 adults over 50 years found that education and employment had modest direct effects on the trait “Openness to Experience”, with no significant gender differences ([24]). Several studies have investigated this relationship further ([40]). For example, [35] ([35]) explored the association between “Openness to Experience” and educational attainment, suggesting that the observed link might be influenced by self-perception bias: more educated individuals tend to overstate their “Openness” in self-report measures, as they place relatively high value on this trait. In our sample of women who sought a VTP, “Openness to Experience” was higher among those with higher education levels. Conversely, women with lower educational attainment showed lower scores in the “Agreeableness” domain.

Many studies shed light on the associations between personality traits and socio-demographic factors and attempt to show which circumstances may influence women’s decisions regarding voluntary termination of pregnancy ([31]). However, this interpretation should be contextualised. We know that pre-existing personality traits may influence both educational trajectories and reproductive choices ([1]; [27]). In this case, we cannot make any claims about the direction of this association.

In our study we administered the PID-5 (Personality Inventory for DSM-5) in order to assess whether any participant presented a dysfunctional personality profile in any of its domains. The PID-5 is a tool developed to evaluate maladaptive personality traits ([23]) across five higher-order domains: Negative Affectivity, Detachment, Antagonism, Disinhibition, and Psychoticism. Several studies have established its construct validity and reliability, both in clinical and community samples.

According to [49] ([49]), individuals who display lower scores across the PID-5 domains are more likely to belong to a “high adaptation/low symptomatology” group, which may correspond to less maladaptive personality functioning ([49]). In the literature, few studies have employed the PID-5 to assess personality in women who sought a voluntary termination of pregnancy (VTP). Previous research has primarily focused on mental health outcomes following abortion or miscarriage, rather than on mental health status prior to undergoing VTP ([9]; [19]; [51]). Notably, [48] ([48]) evaluated pre-abortion mental health, specifically symptoms of depression and anxiety, and found that many of these symptoms were linked to perceived stigma ([48]).

Although some women displayed relatively higher scores in certain BFQ (Big Five Questionnaire) domains (which are intended to measure normal personality traits), these did not correspond with pathological levels on the PID-5. Consequently, we judged these BFQ variations as within non-pathological range for all participants. PID-5 scores showed negative correlations with almost all domains of the BFQ personality traits, except for “Extraversion” and “Emotional Stability”, which had positive but non-statistically significant correlations with “Antagonism”. It should be noted, however, that among the PID-5 dimensions, the participants exhibited moderately higher “Negative Affectivity” compared with the other domains of the PID-5, albeit still within normal values. In this study, the “Negative Affectivity” scores were lower for women with children compared with those without. In the literature, we did not find any studies that investigated the differences in the PID-5 questionnaire with parity, but Carmona-Monge et al. observed a significant relationship between negative auto-focused coping and number of children in relation to pregnancy-related worries ([12]). In [30] ([30]), the study demonstrated that “Disinhibition” was associated with education level ([30]). “Disinhibition” scores were higher among women with lower educational attainment compared with those with high educational attainment. “Detachment” was significantly associated with a higher score in women with lower and medium educational attainment compared to those with high educational attainment. Finally, “Psychoticism” was more pronounced in women with lower education levels compared to those with higher education levels.

With the GDMS, no single predominant decision-making style emerged among the participants; rather, a combination of all styles was used to confront such an important and delicate decision, confirming findings from other research on decision-making, including studies involving women who had undergone VTP ([15]). Regarding the relationship between personality traits and decision-making styles, in our sample of women who decided to undergo VTP, a significant association was found between a high level of “Conscientiousness” and a “Rational” decision-making style. More conscientious women tend to use a more rational decision-making style, a finding that aligns with previous research by [17] ([17]) among medical students in Lebanese universities ([17]). Similarly, those who exhibit higher levels of “Disinhibition” tend to have lower scores in the “Rational” dimension. Women with lower scores in “Conscientiousness”, who also exhibit lower “Emotional Stability” and higher “Extraversion”, tend to make decisions with a “Spontaneous” style, as showed before in other studies.([5]; [6]; [17]). Lower “Conscientiousness” and lower “Emotional stability” are traits that distinguish women who adopt an “Avoidant” decision-making style. Therefore, women who are more extroverted and/or less emotionally stable and less conscientious tend to make decisions more quickly and spontaneously, consistent with observations made by Bayram et al. ([6]). In interpreting the GDMS results, we observed that several PID-5 dimensions were strongly associated with “Avoidant” and “Spontaneous” decision-making styles in univariate models, but these associations did not survive multivariable analysis. Collinearity diagnostics indicated that multicollinearity was not problematic. Rather, this pattern likely reflects overlapping variance among correlated personality dimensions: associations that appeared in univariate analyses were accounted for by other predictors once included together in the multivariable models. These results highlight the importance of considering adjusted models as more conservative estimates of independent effects.

Our findings also have implications for clinical counselling. Although psychometric questionnaires are not routinely used in VTP settings, the observed variability in personality traits and decision-making styles suggests that women approaching VTP do not constitute a homogeneous group. This heterogeneity indicates that a “one-size-fits-all” counselling model may not adequately meet their diverse needs. For example, “Avoidant” or “Spontaneous” decision-making styles may require empathetic guidance to reduce stress and build confidence, while more conscientious profiles may benefit from structured, information-rich communication. Tailoring counselling approaches to different socio-demographic backgrounds, such as education or parity, may further improve the effectiveness and perceived quality of care.

Beyond the individual level, these results should be interpreted within broader theoretical frameworks. Research has highlighted the role of stigma, socio-economic inequalities, and cultural and religious norms in shaping women’s reproductive choices ([38]; [47]). In Southern Italy, where Catholic values continue to exert influence, cultural pressures may intensify decisional stress and shape how individual traits are expressed. Thus, psychological factors should be understood as interacting with structural and cultural dimensions rather than as isolated determinants.

This study has some limitations. First, the dataset did not include measures of religiosity, which represents a key factor in reproductive decision-making in this context. Second, a possible selection bias cannot be excluded: women with greater social resources or lower perceived stigma may have been more willing to participate, potentially underrepresenting more vulnerable groups. Third, the monocentric nature of the study and the limited representation of foreign women and surgical VTP procedures, compared with national data from the Italian Ministry of Health ([41]), may limit generalizability of the findings.

Moreover, other relevant aspects—such as social and cultural variables not captured by the BFQ, PID-5, or GDMS (e.g., neurodivergence), gender expectations within a patriarchal society, and experiences of women who have undergone abortions in illegal or stigmatised contexts—were not included in the present study. Future research should therefore integrate cultural and religious variables and recruit more diverse populations to deepen our understanding of how individual, social, and structural factors jointly influence VTP decision-making.

A strength of this study is that, for the first time, the associations between personality traits, decision-making styles, and socio-demographic characteristics for women who have undergone VTP were assessed through the BFQ, PID-5, and GDMS questionnaires. Furthermore, another strength is the accuracy in administering the psychological questionnaires, which allowed us to obtain reliable data with minimal missing responses, enabling us to present a well-defined and undistorted picture of the psychological traits characteristic of women undergoing VTP.

## 5. Conclusions

This study has delineated a personality profile for women who have had an abortion at the Family Planning Unit of the “Di Venere-Fallacara” hospital in Triggiano, Puglia, Italy, between November 2023 and January 2024. The high average scores in the BFQ traits, particularly “Conscientiousness” and “Openness”, indicate that personality dimensions vary among women based on factors such as employment and education. Employed women exhibit greater “Extraversion”, while higher education correlates with increased “Openness”, and lower “Agreeableness” is observed in those with less education. The dimensions of “Disinhibition”, “Detachment”, and “Psychoticism”, as measured by the PID-5 questionnaire, are also significantly influenced by education level, with medium and low educational attainment impacting these traits and accentuating them. The participants utilised a combination of decision-making styles when confronting the significant decision of having an abortion. Our findings suggest that these women exhibited a more “Rational” decision-making style, accompanied by higher levels of “Conscientiousness” and lower tendencies toward “Avoidant” and “Spontaneous” styles.

## Figures and Tables

**Figure 1 ejihpe-15-00214-f001:**
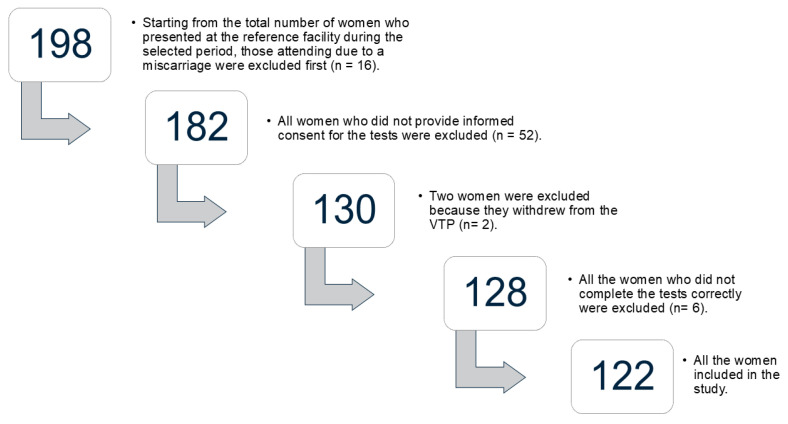
Flowchart of the selection process of the women that participate at the study.

**Figure 2 ejihpe-15-00214-f002:**
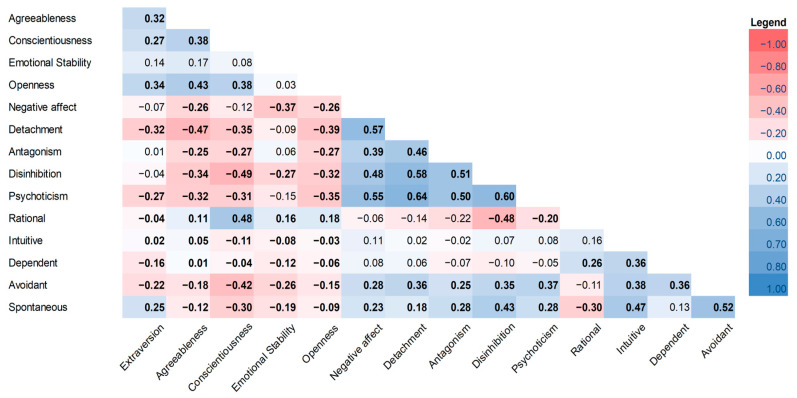
Correlation matrix between and within the BFQ, PID-5 and GDMS tests. Stronger positive correlations are highlighted in blue, while stronger negative correlations are highlighted in red. Statistically significant coefficients (*p* < 0.05) are in bold. Note: In this correlation table, the upper triangle is omitted intentionally to avoid redundancy, and symmetric entries are not repeated. Thus, the last items (e.g., “Extraversion”, “Spontaneous”) are not explicitly shown along both axes but are implicitly included in the matrix.

**Figure 3 ejihpe-15-00214-f003:**
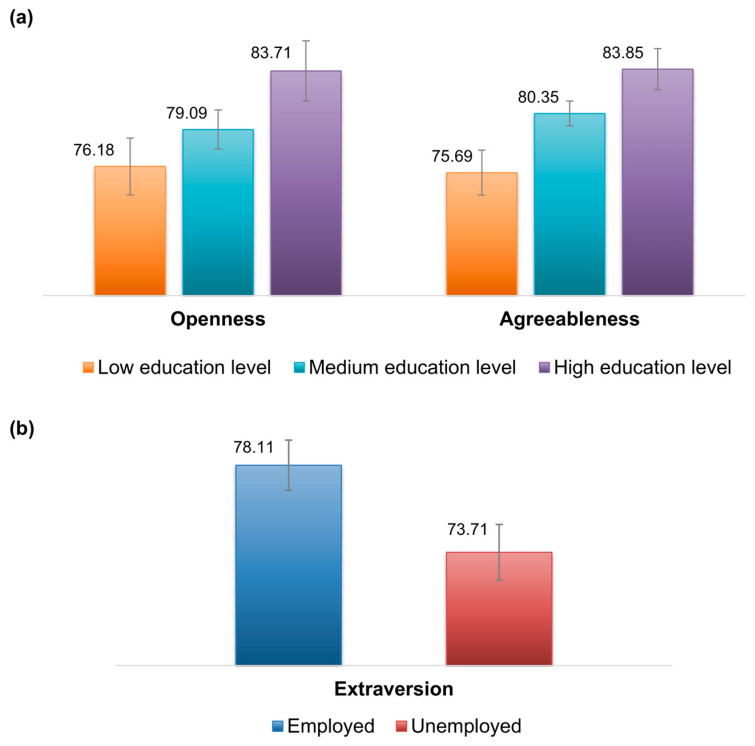
Results of the multivariate analysis of covariance comparing BFQ scores and socio-demographic characteristics. (**a**) Estimated mean scores on the BFQ dimensions “Openness” and “Agreeableness” by education level. (**b**) Estimated mean score on the BFQ “Extraversion” by employment status.

**Figure 4 ejihpe-15-00214-f004:**
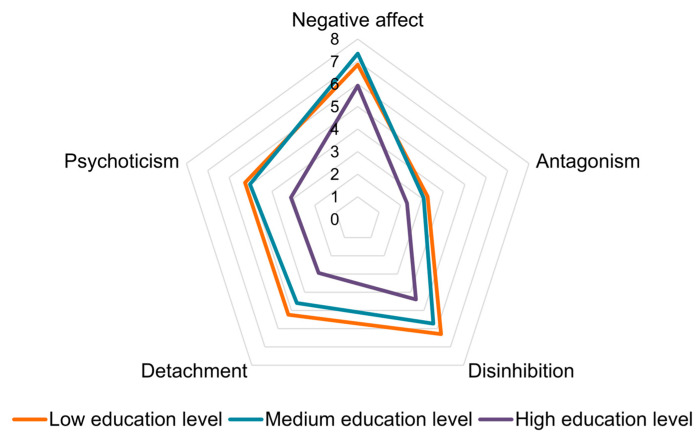
Results of the multivariable analysis of covariance comparing PID-5 scores and socio-demographic characteristics. Estimated mean scores on the five dimensions of the PID-5 by education level.

**Figure 5 ejihpe-15-00214-f005:**
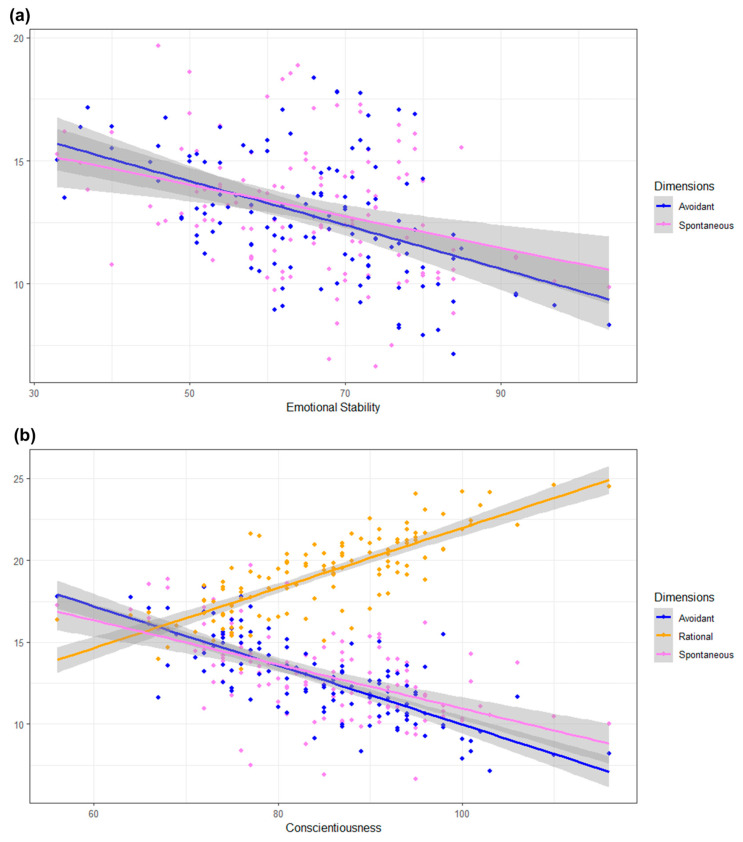
Results of the multivariable analysis comparing GDMS score and socio-demographic characteristics, BFQ, and PID-5 scores. (**a**) Estimated relationship between the PID-5 “Emotional Stability” dimension score and the GDMS “Avoidant” and “Spontaneous” decision-making style scores. (**b**) Estimated relationship between the PID-5 “Conscientiousness” dimension score and the GDMS “Avoidant”, “Rational” and “Spontaneous” decision-making style scores.

**Table 1 ejihpe-15-00214-t001:** Socio-demographics characteristics of women that choose VTP.

Socio-Demographics and VTP’s Related Variables	n (%)
**Age (years)**	
<30 years	48 (39.3)
≥30 years	74 (60.7)
Missing	0 (0)
**Province of residence**	
Bari	96 (78.7)
Outside Bari province	25 (20.5)
Missing	1 (0.8)
**Citizenship**	
Italians	115 (94.3)
Not Italians	7 (5.7)
Missing	0 (0)
**Marital Status**	
In couples	65 (53.3)
Not in couples	55 (45.1)
Missing	2 (1.6)
**Education Status**	
Low educational level	22 (18.0)
Medium educational level	71 (58.2)
High educational level	29 (23.8)
Missing	0 (0)
**Employment Status**	
Employed	66 (54.1)
Unemployed	55 (45.1)
Missing	1 (0.8)
**Live Births (number)**	
0	58 (47.5)
≥1	64 (52.5)
Missing	0 (0)
**Miscarriage (number)**	
0	105 (86.1)
≥1	17 (13.9)
Missing	0 (0)
**VTP (number)**	
0	88 (72.1)
≥1	34 (27.8)
Missing	0 (0)
**Parity**	
With children	64 (52.5)
Without children	36 (29.5)
Missing	0 (0)
**Repeated VTP**	
Yes	34 (27.9)
No	66 (54.1)
Missing	0 (0)
**Gestational Age**	
First 90 days	118 (96.7)
Over 90 days	4 (3.3)
Missing	0 (0)
**Amenorrhea (weeks)**	
≤8	104 (85.2)
≥9	14 (11.5)
Missing	4 (3.3)
**Foetal Malformations**	
Yes	5 (4.1)
No	117 (95.9)
Missing	0 (0)
**Certification Release**	
Family planning counselling facilities	48 (39.3)
Family Doctor	23 (18.9)
Obstetrics-Gynaecology Service of the Healthcare Institute	51 (41.8)
Missing	0 (0)
**Urgency**	
Yes	32 (26.2)
No	90 (73.8)
Missing	0 (0)
**Type of VTP intervention**	
Surgical	1 (0.8)
Pharmacological	119 (97.5)
Missing	2 (1.6)
**Antalgic Therapy**	
General Anaesthesia	3 (2.5)
Analgesia without Anaesthesia	12 (9.8)
No Therapy	105 (86.1)
Missing	2 (1.6)
**Period between certification and VTP**	
≤14 days	121 (99.2)
≥15 days	1 (0.8)
Missing	0 (0)

VTP, Voluntary Termination of Pregnancy.

**Table 2 ejihpe-15-00214-t002:** Psychological test scores of women who chose VTP.

Test	Dimension	Score		
		N	Min–Max	Mean (SD) or Median [IQR]
**BFQ**	Extraversion	122	47–106	76.11 (9.58)
	Agreeableness	122	58–102	80.34 (8.56)
	Conscientiousness	122	56–116	84.49 (10.50)
	Emotional Stability	122	33–104	65.47 (13.07)
	Openness	122	52–109	80.61 (10.42)
**PID-5**	Negative affect	122	0–15	7 [5–9]
	Detachment	122	0–13	4 [2–6]
	Antagonism	122	0–12	2 [1–5]
	Disinhibition	122	0–13	5 [2–7]
	Psychoticism	122	0–14	4 [2–7]
**GDMS**	Rational	122	9–25	19 [16–23]
	Intuitive	122	8–25	18.12 (3.35)
	Dependent	122	5–25	16 [14–19]
	Avoidant	122	5–25	12 [10–16]
	Spontaneous	122	5–25	13 [9–16]

BFQ, Big Five Questionnaire; PID-5, Personality Inventory for DSM-5; GDMS, General Decision-Making Style.

## Data Availability

The data supporting the findings of this study are not publicly available due to privacy and ethical restrictions. Interested researchers can contact the corresponding author for further details regarding data access and usage, subject to approval.

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
