# Peer review of "Association Between Decision-Making Styles, Personality Traits, and Socio-Demographic Factors in Women Choosing Voluntary Pregnancy Termination: A Cross-Sectional Study"

_ejihpe, 2025, doi:10.3390/ejihpe15100214_

Round 1
Reviewer 1 Report
Comments and Suggestions for Authors
This study explores the relationship between personality traits, decision-making styles, and socio-demographic characteristics in women seeking voluntary termination of pregnancy (VTP) in Southern Italy. Using validated instruments—the Big Five Questionnaire, the Personality Inventory for DSM-5 (PID-5), and the General Decision-Making Style scale—the researchers analyzed data from women recruited at a monocentric hospital. The findings show generally high levels of conscientiousness and openness, with no evidence of personality dysfunction, and associations between certain socio-demographic factors and decision-making styles. The study concludes that women seeking VTP represent a psychologically heterogeneous group and highlights the potential value of considering personality and decision-making profiles in providing more tailored counseling and support.
Strengths:
This paper addresses an important and underexplored question: how personality traits, decision-making styles, and socio-demographic factors interact in women undergoing voluntary termination of pregnancy. The topic is highly relevant to women’s health psychology, and the authors clearly invested significant effort in collecting and analyzing a well-defined dataset. The writing is clear, the methodology transparent, and the statistical analysis well-documented. However, the paper remains somewhat descriptive and surface-level in its current form, and its framing limits the depth and impact of its contribution.
Here are 4 points I suggest for improving the manuscript:
- Religiosity and cultural context: The manuscript would benefit from considering the potential role of religious beliefs in shaping women’s experiences and decision-making regarding voluntary termination of pregnancy. Given the cultural and social context of Southern Italy, where religion may exert a significant influence on reproductive attitudes and perceived stigma, the absence of data on this variable limits the depth of the analysis and interpretation. At a minimum, acknowledging this limitation in the discussion would strengthen the paper and highlight an important avenue for future research.
- From findings to practice: The paper promises to inform counseling interventions but does not translate findings into actionable recommendations. While the discussion briefly mentions the potential value of tailoring counseling and support based on women’s personality and decision-making profiles, this point remains underdeveloped. To enhance the practical relevance of the study, it would be helpful to expand on how these findings could inform clinical practice or counseling strategies. For example, the authors could suggest how certain traits or decision-making styles might guide communication approaches, support planning, or interventions aimed at reducing stress and improving decision satisfaction. Providing such insights would strengthen the applied significance of the research.
- Contextualizing personality traits: The discussion could be strengthened by clarifying whether the personality traits identified—such as higher conscientiousness and openness—are specific to women seeking voluntary termination of pregnancy or instead reflect broader life circumstances and resources, such as higher education, access to information, and supportive social networks. Additionally, the authors should consider the potential for selection bias: women with more personal and social resources may have been more willing to participate in the study, while those facing greater stigma or socio-demographic barriers may have been underrepresented. Expanding on this point would help contextualize the findings and avoid overgeneralization.
- Deeper theoretical engagement: The manuscript would benefit from deeper engagement with the broader literature on abortion decision-making, particularly in relation to stigma, structural inequities, and the cultural and religious context of Southern Italy. At present, the discussion remains primarily descriptive, which limits the theoretical depth of the paper and makes it read more like a dataset report than a critical psychological analysis. Integrating current debates on how external stressors and structural factors interact with individual traits could enrich the interpretation of the findings and enhance the paper’s contribution to the field.
- In line 286 of the discussion, the manuscript refers to women being “interviewed,” but based on the methods section, it appears that data were collected exclusively through self-administered questionnaires. Clarifying this wording to avoid confusion would improve accuracy and transparency.
Author Response
We sincerely thank Reviewer for the time dedicated to reviewing our manuscript and for the constructive comments and suggestions provided. We have carefully considered each point and made the corresponding revisions to improve the quality and clarity of the paper. Below, we provide a detailed, point-by-point response to all comments.

Reviewer 2 Report
Comments and Suggestions for Authors
The theme is interesting, due to the sample (women performing VTP) not sufficiently studied; however, the relation among the variables, the chosen data analysis and many incorrect statements, with conclusions that, in my opinion, may not be extracted from their study. Beginning in the abstract and ending in the discussions, leads me to consider this article lacks quality. Pertaining to Data analysis, I don´t understand the use of Regression analysis, with sociodemographic variables used as predictor of personality variables..conceptually, I don´t think it makes any sense, I think Analysis of Variance (ANOVA/MANOVA) analyzing the difference in the mean values of personality according to different socioeconomic level, employment level, etc. The Big Five variables are traits, endogenous dispositions with a strong biological base that are relatively stable; we know that throughout life their levels may be affected by life events, but he strongest relation is the reverse: different levels in the personality traits lead persons to different choices and paths is life, as different school paths, employment success vs. unsuccess, stability vs. instability in relationships…in p.11 line 310 the authors state “A greater degree of "Extraversion" was observed among women with stable employment compared to unemployed women, confirming by De Vries et al. (2021) (De Vries et al., 2021) suggestion about the importance of work in reinforcing extraversion traits. Similarly, "Openness to Experience" was higher among women with a higher education level, consistent with personality development models related to education (Ludeke, 2014; Verbree et al., 2023). Lower "Agreeableness" was found for women with lower level of education. These results align with previous findings that highlighted how socio-demographic circumstances influence not only personality traits but also women's choices regarding VTP (Coast et al., 2018; Idris et al., 2023; Kassa et al., 2024; Sorhaindo & Lavelanet, 2022). By the aforementioned, this interpretation should be contextualized and the opposite interpretation should be presented. Pertaining to the last part of the statement, on the influence of socio-demographic circumstances on the women's choices regarding VTP, it does not make sense because the present study did not explore this issue. It did not compare any profile of women performing vs. non performing VTP, in which case this kind of statement could make sense.
In the same vein, in p.11, line 292: “As found by Ferreira et al. (Ferreira et al., 2015b), our study also clearly confirms that 292 the decision to resort to VTP depends on various situations determined by a person’s so-293 cial context. Social factors, such as employment, being in a relationship, having a medium 294 level of education, and having children, more frequently affected women who decided to terminate their pregnancies, confirming the results highlighted in previous studies (Biggs 296 et al., 2013;. – I think the authors cannot have this interpretation from their results, as they did not compare women performing VTP with women not performing VTP.
And from this I present my next idea: It would be much more interesting to study differences in the variables of interest between different groups of women in the sample (e.g., by number of VTP, performing VTP for medical reasons vs. with no medical reasons, etc).
In the Abstract:
“Personality traits, decision-making styles, and socio-demographic information help understand the characteristics of women requesting voluntary termination of pregnancy (VTP) and monitor the timing and quality of this service. However, current data do not allow for an in-depth understanding, as other less evident factors may influence the decision.- This statement does not make sense, it seems to indicate that this study Is addressing these “less evident factors”.
Next, “The aim was to describe the VTP process and its relationship with decision-making style, psychological and socio-demographic factors”- this study does not describe the VTP process. I think the only accurate aim could be to characterize a sample of women performing VTP concerning socio-demographic, personality and decision-making variables, and the relation among them.
In the Discussion:
p.11, line 299: no anomalies were found in the personality profiles, either in terms of personality traits or dysfunctional personality characteristics. The average scores obtained through the Big Five personality test are generally high for all traits- The two statements are not coherent with each other, as high values in some Big Five variables would be dysfunctional.
p.11, line 324: Therefore, overall, our research perhaps refutes the suggestion that women requesting VTP live in socio-demographic contexts that force or coerce them to resort to voluntary abortion.- what data in this study allow this conclusion? I can´t identify it.
There several other issues, but these are the main I present to justify my decision.

Author Response

(The authors gave the same response as above.)

Reviewer 3 Report
Comments and Suggestions for Authors
The paper aims to investigate the relationships between socio-demographic factors, personality traits, and decision-making styles in a population of women seeking abortion at a single medical center. Collecting this amount of data is a significant undertaking, and the authors did a good job here. The analysis plan is sound and generally carried out correctly, with a few minor omissions. The results are compelling and important. However, major revisions are required.
GENERAL COMMENTS
The study does not compare the sociodemographic profile of women seeking VTP with the socioeconomic profile of the population in the region from which the subjects are drawn. Therefore, no claims about frequency can be made, nor can predictions based on the sociodemographic data be made. This is a critical flaw in the interpretation of the data that must be corrected. Either the conclusions must be adjusted, or population data must be collected, analyzed, and shown to support the claims of frequency for sociodemographic characteristics.
There is a large loss of significance between univariate and multivariate analyses. This implies strong collinearity in some of the parameters. This should be analyzed and discussed, rather than reporting results considered significant at p<0.25 in univariate analysis and saying that the significance disappeared in multivariate analysis without explanation. Discussion of univariate results is distracting and dilutes the overall conclusions. It would probably be more effective to get the collinearity out of the way, discuss the multivariate results, and draw conclusions from those results. In addition, Figure 5 and lines 344-359 of the conclusions (relationships between PID-5, GDMS and BFQ) should be set in the context of previous findings more firmly. In other words, these findings are true for the general population, not just women seeking VTP. It would be more accurate to state that the women in this study display the expected relationships between these variables.
ABSTRACT
Lines 29-30: It should be noted that the levels of Detachment and Pyschoticism were not pathological (as in line 222-223).
Lines 30-31: Specific data should be added
MATERIALS AND METHODS
Line 196: The p<0.25 cutoff may not be familiar to most people. Please add a reference (e.g. Hosmer, Lemeshow, et al.), and add a statement that this is standard for multivariate regression analysis to find associations that may not be evident in univariate analysis with the traditional p<0.05 cutoff.
RESULTS
Table 1. Many of the metrics do not add up to 122. If there is missing data, it should be listed and included in the percentages. Percentages on less than 122 is not appropriate (e.g., Occupation). In addition, the percentages for Marital Status add up to 90% for 120 individuals. This is incorrect whether or not the missing two individuals are considered as part of the total. “Occupation” should be replaced with “Employment Status”, and “Born Alive” should be replaced with a better term such as “Parity” or “Live Births”.
Figure 2. The term “Emotional Stability” is used in the figure, while “Neuroticism” is used in elsewhere in the manuscript (text). One term should be chosen and used consistently throughout the manuscript. “Extraversion” is missing from the y axis and “Spontaneous” from the x axis, without explanation. This should be corrected.
Line 257: The phrase “was lost in the multivariate analysis” should be changed. The term “lost” implies a flaw in the analysis rather than revealing a nonsignificant result. A better phrase would be something like "did not survive multivariate analysis.”
Line 260-262: These should be removed.
Line 265 (Figure 3 caption): “Occupation” should be replaced with “Employment Status”.
Lines 292-293: The following statement overinterprets the data of this study: “As found by Ferreira et al. (Ferreira et al., 2015b), our study also clearly confirms that the decision to resort to VTP depends on various situations determined by a person’s social context.” Neither this manuscript nor Ferreira et al., 25015b examines how a person’s social context affects the decision to resort to VTP. Both studies looked at the attributes of women who sought or received VTP without comparing the results to population attributes or women who did not seek VTP. Neither can be considered to contain predictive information about how social context will affect the decision to seek VTP.
294-296: The following statement overinterprets the data of this study: “Social factors, such as employment, being in a relationship, having a medium level of education, and having children, more frequently affected women who decided to terminate their pregnancies.” More frequently than whom? The only thing that can be said is that the majority of women who decided to terminate their pregnancies at this center were employed, in a relationship, had a medium education level, and had children.
Figure 5: The data underlying Figure 5 is not discussed in Results section, nor is Figure 5. It feels like at least one paragraph was removed accidentally. Please add appropriate text to Results section.
Line 327: There is no comparison group noted. This should be corrected. In other words, moderately higher “Negative Affectivity” compared to what group?
Line 332-333: This was not statistically significant in multivariate analysis.
Lines 344-348: The authors state that their findings are confirmed by Othman et al. 2020. This is untrue. The findings in this manuscript confirm those previously made by Othman. Please rephrase.
REFERENCES
The following references returned website results of Error 404 and could not be found elsewhere for evaluation:
- n. 194/1978., 1978
EpiCentro, 2022
GU No. 109, 12-05-2017, 2017
The following references do not support the statements in the manuscript and must be removed or replaced, or statements must be amended:
Zajenkowska et al., 66 2022. This study does not explore decision making in the context of abortion.
Idris et al., 2023. This review article examines women’s healthcare decisions, not reproductive decisions specifically. Either cite more exact source material or broaden the claim that uses this citation.
Reardon et al., 2023. This article examines how “rightness” of a decision to abort affects emotion, mental health, etc. It does not support the statement “individual differences in cognitive style and personality traits may influence how decisions about abortion are experienced and justified.”
Adesse et al., 2016. This review analyses the effect of the stigma associated with abortion on women. It does not support the statement regarding the limitations of studies examining personality traits in women seeking VTP. NOTE: I’m not certain that the statement about the limitations of the current literature needs to be supported.
Gambetti et al., 2008. Article cites Cronbach’s alpha scores of 0.62 to 0.94. Removed citation or amend the statement it supports “prior validation studies reporting coefficients between 0.70 and 0.86.”
Ferreira et al., 25015b. The study does not support the statement “the decision to resort to VTP depends on various situations determined by a person’s social context.” The cited study examined the attributes of women who sought or received VTP, without comparing the results to population attributes or women who did not seek VTP. Therefore, it does not contain predictive information about how social context will affect the decision to seek VTP.
Broen et al., 2005; Fergusson et al., 2006; Wallin Lundell. These studies examined mental health after abortion or miscarriage, not before VTP. They do not support the statement “unlike other studies (Broen et al., 2005; Fergusson et al., 2006; Steinberg et al., 2016; Wallin Lundell et al., 2017), no anomalies were found in the personality profiles, either in terms of personality traits or dysfunctional personality characteristics.”
Steinberg et al., 2016. This study assessed depression symptoms and anxiety symptoms (etc.) in pre-abortion mental health, much of which was attributed to stigma. It does not support the statement “unlike other studies (Broen et al., 2005; Fergusson et al., 2006; Steinberg et al., 2016; Wallin Lundell et al., 2017), no anomalies were found in the personality profiles, either in terms of personality traits or dysfunctional personality characteristics.”
De Vries et al., 2021. This article examines personality traits in the context of two life events: graduating high school and moving away from home. Employment was not examined. This reference does not support the statement “A greater degree of "Extraversion" was observed among women with stable employment compared to unemployed women, confirming by De Vries et al. (2021) (De Vries et al., 2021) suggestion about the importance of work in reinforcing extraversion traits.”
Ludeke, 2014. This article examines the role of biased responding underlying the association between education and “Openness”. It does not support the statement “Similarly, "Openness to Experience" was higher among women with a higher education level, consistent with personality development models related to education (Ludeke, 2014; Verbree et al., 2023).”
Verbree et al., 2023. This manuscript does not mention “Openness” even once. It does not support the statement “Similarly, "Openness to Experience" was higher among women with a higher education level, consistent with personality development models related to education (Ludeke, 2014; Verbree et al., 2023).”
Coast et al., 2018. This is a conceptual framework paper. It does not support the statement “These results align with previous findings that highlighted how socio-demographic circumstances influence not only personality traits but also women's choices regarding VTP.”
Sorhaindo & Lavelanet, 2022. This manuscript examines the effect of stigma on quality abortion care, not women’s decision to obtain VTP. It does not support the statement “These results align with previous findings that highlighted how socio-demographic circumstances influence not only personality traits but also women's choices regarding VTP.”
Stover et al., 2019. This article examines the relationship between PID-5 and the maladaptive personality traits model. While the article includes sociodemographic data, that does not make this a paper that supports the statement “all the women scored low or medium in the test dimensions (Negative affect, Detachment, Disinhibition, Antagonism, and Psychoticism), which are also attributable to critical socio-demographic conditions.”
Jonason et al., 2017. In this study, Disinhibition but not Psychoticism was associated with level of eduction. It does not support the statement “Finally, "Psychoticism" was more pronounced for women with a lower level 335 of education.”
May need to be removed or replaced:
Fernández-Basanta et al., 2023. Could not be evaluated, but the freely accessible information does not support the statement “individual differences in cognitive style and personality traits may influence how decisions about abortion are experienced and justified.”
Could not be evaluated:
Di Fabio, A., 2007.
Cheshure 323 & Lehtman, 2020. This is the original paper introducing the PID-5. It is unlikely to support the statement “Regarding the PID-5, an evaluation of the scores obtained by the subjects in our sample did not reveal any personality dysfunctions: all the women scored low or medium in the test dimensions (Negative affect, Detachment, Disinhibition, Antagonism, and Psychoticism), which are also attributable to critical socio-demographic conditions.”
Kero & Lalos, 2000.
References not evaluated because the statement they support is incorrect and needs to be removed or amended:
Ferreira et al., 2015b; Biggs 296 et al., 2013; Fernández-Basanta et al., 2023; Kassa et al., 2024; Reardon et al., 2023; Sedgh 297 et al., 2012
SUPPLEMENTARY MATERIALS
Tables S1-S3: Regarding Education Status analysis, there should be separate significance tests for Low vs High and Medium vs High. In addition, the Low vs Medium comparisons should be made for all metrics.
Table S3: The first statement should specify univariate analysis. The following statement is incorrect: “The increases in all the PID-5 dimensions were correlated with a decrease in the “Rational” scores, a decrease that was not statistically significant, except for the dimensions “Negative affect” and “Detachment”…” The data show that the decrease was significantly different in the univariate analysis except for “Negative affect” and “Detachment”.
Table S1-S3: A discussion of univariate analysis needs to be followed by a discussion of the multivariate analysis, which makes many of the univariate results moot.
Author Response

(The authors gave the same response as above.)

Round 2
Reviewer 2 Report
Comments and Suggestions for Authors
I think the authors properly addressed my comments and concerns.
Reviewer 3 Report
Comments and Suggestions for Authors
Thank you for completing the requested revisions. I hope you agree that the manuscript is much improved and ready for publication!